# United by Contagion: How Can China Improve Its Capabilities of Port Infectious Disease Prevention and Control?

**DOI:** 10.3390/healthcare10081359

**Published:** 2022-07-22

**Authors:** Danzi Liao, Tianyue Lyu, Jia Li

**Affiliations:** 1College of Public Administration, Zhejiang University of Finance and Economics, Hangzhou 310086, China; cleverdz@foxmail.com; 2School of International Studies, Zhejiang University, Hangzhou 310086, China; lty_terrence@163.com

**Keywords:** post-pandemic era, infectious diseases at port, united by contagion, politicization of epidemic diseases, health and quarantine

## Abstract

The rapid development of the social economy and science and technology has led to more frequent transnational movements of people, goods and vehicles. At the same time, various cross-border risks have significantly increased. The rapid global spread and continuous mutation of Coronavirus Disease 2019 (COVID-19) have again exposed the international community’s extreme vulnerability to major transnational public health emergencies. China started a “war against the epidemic” with tight quarantine regulations and border restrictions on people, vehicles and international goods. However, it also revealed the weaknesses in and incapacity for disease prevention and control at ports in terms of obstructed performance of the whole chain of public agencies, incompatible laws and regulations, lack of key technologies, and difficulties in international cooperation. Combined with persuasive data, this paper systematically illustrates how transnational infectious diseases lead humans to be “united by contagion”. On this basis, this paper makes a targeted analysis of the deficiencies of port epidemic prevention and control in China’s fight against COVID-19 and suggests corresponding countermeasures and reflections.

## 1. Introduction: Phenomenon and Problem

The rapid spread of COVID-19 around the world for nearly two and a half years has caused incalculable trauma to the entire human race. Some observers call it “World War III” [1]. According to the Worldometer, as of 1 May 2022, there were more than 500 million confirmed COVID-19 cases and more than 6.26 million deaths worldwide. Moreover, the COVID-19 crisis, with its unprecedented spread scale, speed, damage and insufficient response, has brought profound disruption to global economic and social stability.

The term “port” refers to the gateways and passages of the country’s opening to the outside world and international communication, formed by port, airport, station, wharf, cross-border channel, etc., where people, goods, articles and vehicles directly enter and exit. By the end of 2020, the number of China’s ports has reached 313, with roughly 22.2 billion tons of import and export cargo volume, 146.37 trillion RMB yuan of import and export value, 2.627 billion people and 147 million vehicles [2]. Over the past ten years, the trade value of import and export goods and the number of people entering and exiting China have been the highest in the world. Meanwhile, the coastal, border and inland ports have formed a “security network” for China. As of 20 February 2022, a total of 13,472 confirmed cases of COVID-19 were reported to have imported from outside China. “External anti-input” (prevent the epidemic from entering China at ports) and “internal anti-rebound” (prevent the spread of the domestic epidemic) together form a “two-sided battle” in China’s “war against the epidemic”. Similarly, the Ebola outbreak also led to a similar phenomenon, which showed the absolute need for policymakers and experts in public health to strengthen all the highlighted weaknesses in the management of Ebola, primarily to address the needs of the populations living in the most affected countries [3].

Since 2021, more and more Western countries have begun to treat COVID-19 as a common epidemic and have comprehensively loosened domestic intervention and entry and exit controls. However, China is still maintaining the mode of highly intensive “anti-epidemic war”. Therefore, a direct and obvious question is, what impact will the post-epidemic era brought by the full unblocking of the West have on China’s port epidemic prevention? How should the evolution of infectious diseases at ports be viewed in the post-epidemic era? What theoretical explorations have been made by scholars, what are the shortcomings in China ports, and what should be the corresponding improvement plans for the prevention and control capacity of infectious diseases at ports? This paper will try to answer these questions.

## 2. United by Contagion: Why Is It Urgent and Vital to Prevent and Control Infectious Diseases at Ports?

Diseases do not recognize national borders. COVID-19 again proved that the world is “united by contagion” [4]. Therefore, all of humanity is “connected by risk”.

First, new emerging and reemerging infectious diseases are increasing with intensified risks globally. According to statistics, more than 1000 infectious diseases have been identified worldwide and 360 have been officially reported in China [5]. The international community has confirmed more than 250 zoonotic infectious diseases, with roughly 130 confirmed in China [6]. According to the World Health Organization (WHO), more than 80% of the global population is at risk of vector-borne disease [7]. According to a Nature article published in 2019, at least 150 pathogens that affect humans have been identified as emerging, re-emerging or evolving since the 1980s, and ancient diseases such as tuberculosis, dengue fever, malaria, Caribbean chikungunya, Middle East respiratory syndrome (MERS) and Ebola are resurgent and even incurable [8]. Over the past 50 years, more than 40 new infectious diseases have emerged globally and are still keep increasing at a rate of 1–2 per year, with more than 20 identified in China [9]. Subsequently, global governance capacity appears somewhat inadequate.

According to WHO epidemic reports, 70 countries or regions worldwide have had Zika virus disease epidemics since 2007, with 52 countries or regions with locally transmitted cases. China has reported a cumulative total of 30 cases of imported Zika virus disease from 2016–2019 [10]. The severity of the consequences of the Zika virus has prompted WHO to issue the document, Zika: Strategic Response and Joint Operations Plan [11]. Monitored data show a significant increase at the Chinese border: the confirmed cases of infectious diseases witnessed an increase of nearly 80% in 2018 compared with 2015, and persons with symptoms of infectious diseases in 2018 increased by nearly 22.5% compared to 2015 (Table 1). At the beginning of the 21st century and especially in the last decade or so, infectious diseases passing through the Chinese border have become increasingly diverse, with epidemics such as malaria, AIDS, COVID-19, SARS, Ebola, MERS, Influenza A (H1N1), highly pathogenic H5N1 avian influenza, human H7N9 avian flu, African swine fever, Mongolian plague and other significant outbreaks of acute infectious diseases.

The mass spread of infectious diseases brings enormous threats to life, health, economy and society. As of 1 May 2022, the number of COVID-19 infections that have not yet been brought under absolute control has exceeded 500 million and deaths reached 6.26 million. Influenced by the pandemic, the 17 UN Sustainable Development Goals, including reducing inequality, reducing carbon emissions and fighting hunger, have stalled or gone into reverse: the number of people falling into poverty in 2020 increased by 119 to 124 million, the number of the starving population rose from 83 to 132 million, 255 million full-time jobs were lost and 101 million more children and young people did not reach the minimum reading proficiency level [12].

The 1918 influenza pandemic was the most severe in the 20th century. It is estimated by the American Centers for Disease Control and Prevention (CDC) that about 500 million people, or one-third of the world’s population, became infected with this virus, and the number of deaths was estimated to be at least 50 million worldwide [13]. According to WHO, the number of deaths from all infectious diseases in the 20th century (1.4 billion) was nearly 14 times greater than the number of deaths from all wars (110 million) in the same period [14]. In 1990, on the eve of the end of the Cold War, although the threat of nuclear war was basically lifted, the number of deaths due to infectious diseases accounted for 16.69 million (34.4%), nearly 52 times the number of deaths due to war (322,000, accounting for 0.64%) [15]. Additionally, large-scale epidemic events can cause regional and global panic and social disorder. COVID-19 has led to successive ethnic conflicts and violence in several countries and regions.

Second, the flow of people, goods, logistics, data, and capital across China’s boundaries has increased the risk of infectious diseases at the China ports. From 2010 to 2019, the number of Chinese and foreign tourists entering and leaving China increased from 382 million to 670 million (97 million of whom were foreign nationals), and the number of cross-border vehicles increased from 23.54 million to nearly 36.24 million vehicles [16]. Corresponding to this sharp increase, the number of infectious diseases at ports has also significantly increased. A total of 1398 falciparum malaria cases were reported in 2011, of which 1366 cases were imported from abroad, accounting for 98% of the total [17]. During the past decades, China’s ports have detected thousands of cases of imported syphilis, AIDS, tuberculosis, dengue fever, chikungunya fever, human avian influenza and other infectious diseases and intercepted millions of mosquitoes, rats and other medical vectors that spread the plague, malaria, hemorrhagic fever and other infectious diseases at the ports. The proportion of infectious diseases detected in people entering and leaving China from countries along the “Belt and Road” increased from 9.9% in 2012 to 33.6% in 2016 [18]. From January to July 2019, the entry quarantine at National Customs detected 12,565 cases of infectious diseases, an increase of 8.62% year on year. In recent years, imported falciparum malaria, cholera, dengue fever, Zika, chikungunya fever, and other malignant infectious diseases have been detected at frontier ports for the first time. In 2016 and 2017, respectively, imported Rift Valley virus (RVF virus) and Ekpoma virus were found in inbound travelers for the first time and the detected Ekpoma virus is the second case in the world.

Additionally, China has frequently detected the new coronavirus from imported cold chain food, international mail packages and live animals. In recent years, several ports in Shanghai, Beijing, Tianjin, Shenzhen and Chengdu have intercepted highly toxic live organisms in inbound international parcels, such as poisonous scorpions, arrow frogs, Brazilian tortoises, poisonous spiders and poisonous snakes. During the period of the 12th Five-Year Plan (2011–2015), nearly 9000 species of exotic pests were intercepted at Chinese ports, with an average annual growth rate of 26.8%, and more than 95% of invasive species were brought into China by people [19]. During the 13th Five-Year Plan period (2016–2020), the cumulative interception of plant pests alone has reached 8858 species and 3.6 million times [20]. In 2020, affected by the epidemic, despite the fact that the total value of imported and exported goods decreased by 1.1% compared with 2019, nearly 70,000 species of harmful organisms were still intercepted in inbound goods, a year-on-year increase of 15%; of the approximately 420,000 animals pre-inspected overseas, approximately 100,000 animals were unqualified for quarantine, a year-on-year increase of 27%, accounting for 31% of the unqualified animals eliminated during overseas pre-inspection during the 13th Five-Year Plan period (2016–2020). In 2019, more than 660 invasive alien species have been found across China, an increase of more than 30% from 10 years ago, causing annual economic losses of more than 200 billion Chinese yuan to China [21]. At the same time, the prevention and control of the risk of nuclear, biological and chemical violence is also an important security task for China’s ports [22]. In short, globalization makes Ebola, SARS, H1N1, COVID-19 and other infectious diseases spread rapidly worldwide with convenient transportation. In other words, globalization has also become the era of global contagion.

Third, international co-management of infectious diseases at ports becomes more difficult with the politicization of contagious diseases. The study found that infectious diseases fundamentally impact the rise and fall of great nations through several factors, including sharp population reductions, civil unrest and increased political uncertainty when a leader is infected [23].

A wave of cholera epidemics across Europe in the 1830s and 1840s catalyzed a new era of ‘infectious disease diplomacy’ globally [8]. In the era of globalization, public health issues are very closely related to international politics and infectious diseases are both contagious and “global governance diseases”, as the German politician Rudolf L. K. Virchow put it: Medicine is a social science and politics is nothing else but medicine on a large scale [4]. 

In 1995, Dennis Pirages, a professor of political science at the University of Nevada, discussed the relationship between public health, infectious disease governance and human well-being and derived a new concept of “microsecurity” from “microorganisms”. It was the first to include infectious diseases in the study of political science [24]. In 1998, Professor David P. Fidler of Indiana University analyzed the connection between viral microbes, infectious disease control and national interests, international politics, the international system and global society and coined a new term for it—“microbialpolitik”. He also considered infectious disease control as “a global political phenomenon and a political process” [25]. Public health issues have been politicized.

The politicization around COVID-19 is apparent: the U.S. and its allies have engaged in intense political and public opinion confrontations with China over international cooperation in combating the epidemic, vaccine development, anti-epidemic measures and virus traceability. For example, China accused former U.S. President Donald Trump of deliberately stigmatizing the term “coronavirus” to “China virus” in his speech. China also interpreted labels, such as the “Wuhan virus” and “sick man of East Asia” in a highly politicized manner, as insults and hatred towards China from the West. The Chinese Ministry of Foreign Affairs website also reveals 24 labels of “fabricated lies” by the West such as “China virus”, “Wuhan virus”, “virus originated from Wuhan lab”, “China deliberately spread the virus to the world” [26]. In addition, when the Western media called China’s entry controls a “denial of human rights”, and when *The New York Times* identified China’s city closures as a “great loss of personal freedom” but interpreted Italy’s similar action as “risking to the containment of an epidemic in Europe”, China was adamant that this was a “double standard” and “political export” by the West. China also interpreted the West’s insistence on going to Wuhan to trace the virus as an “ulterior motive”. For large countries, claims of vaccine development and distribution are also often interpreted as “conspiracies” or “political kidnappings” with a specific political agenda.

Fourth, China has taken three initiatives to improve infectious disease prevention and control capacity at ports. (1) Since the 18th CPC National Congress in 2012, especially the 19th Congress, the Central Committee has put forward the modernization of the national security governance system and capacity at an unprecedented level. The Committee has been focused on the new era of “integrating development and security”, “high-quality development”, “high-level security” and “prevention and resolution of major risks”. Moreover, China has made “adhering to the overall national security concept” one of the 14 essential strategies of Socialist with Chinese Characteristics for the New Era, highlighting the rule of law, strategy, science and technology disciplines and education in national security. (2) China has prioritized strengthening the prevention and control of infectious diseases, incorporating it into the national security strategy and implementing the “Health China 2030” strategy. In particular, in the face of the impact of COVID-19, China takes crucial measures to strengthen the national biosafety risk prevention and control capacity and prevent and control outbreaks at ports. (3) The institutions have been reorganized. In 2018, a new round of national institutional adjustment carried out a comprehensive reform of the port security system: the entry-exit inspection and quarantine functions in the General Administration of Quality Supervision, Inspection and Quarantine were all assigned to the General Administration of Customs, that is, the formation of the new General Administration of Customs. One of the three changes to the functions of the New General Administration of Customs approved by the central government is to “strengthen the supervision of port security”. Therefore, “the scope of customs port supervision has been expanded, and the task of maintaining port security is more difficult and heavier than ever. We must focus on the overall national security situation, strengthen the port security control capabilities, hold the bottom line of supervision and effectively break the security threat” [27].

## 3. Literature Review

The security of infectious diseases at ports is one of the topics of port security, which has been gradually developed by Chinese scholars who are concerned about the security and risks of entry and exit in the context of openness in recent years. It is currently receiving attention from researchers in public affairs, international trade, international economics and politics. Professor Yu Xiaofeng from Zhejiang University defines “port security” as the security related to the entry and exit of people, goods, articles and transportation vehicles [28]. Yu’s book, *From “Port Security” to “Field Security”*, also compares port security with the 11 fields of Overall National Security proposed by China’s President, Xi Jinping, in 2013. The book is essentially a systematic discussion of securing port matters, prompting the theoretical concerns of port threat identification and risk control. Yu’s book also puts forward a new theoretical perspective—field security—for the study of port security, proposing that port security has the characteristics of “tech-related”, “disaster-related” and “foreign affairs-related”. Wang Feiyi and Huang Shengqiang from Shanghai Customs Institute define “port security” as the national security achieved by public agencies during the import and export of people, goods and means of transportation, and also suggest that in the study of national security, port security is still an important area that has not yet attracted the attention of scholars [29].

What are the types of infectious diseases at ports? Yu Xiaofeng identified four major security problems faced by Chinese ports: ecological damage, unsafety of imported food, substandard product quality and public health disease, among which infectious diseases include SARS, mad cow disease, human highly pathogenic avian influenza, Ebola hemorrhagic fever, Marburg virus disease, tuberculosis, malaria and radioactive overloads [28]. Liao Danzi, based on the objective harmfulness of the causative factors, used the Delphi method to clarify the list of non-traditional security threats at ports and established a threat identification matrix covering five categories (ecological environment, public health, product quality, imported food and nuclear–biological–chemical risks) and seven dimensions (problem area, target, impact level, threat attributes, threat origin, threat carrying tools and landmark events) [30]. Jia Yeqing explored the identification and assessment of port non-traditional security risks with the help of fuzzy neural networks and hierarchical analysis [31]. Ruan Guangzhu analyzed the four major non-traditional security issues of border counter-terrorism, cultural security, environmental protection and anti-drug and anti-smuggling at customs ports and accordingly identified eight infectious diseases in quarantine at ports [22]. Liu Jue classified the risk of imported infectious diseases in China into four categories in terms of risk level [32].

How do cross-border movements of elements trigger the cross-border spread of disease? Are border controls useful for controlling the spread of infectious diseases across borders? An article in *Science* shows that the mobility of various factors strongly contributes to the global reach of COVID-19 [33] and that population movements are the main cause of the spread of epidemics [34]. A modeling study of travel restrictions within Europe suggests that although cross-country airline restrictions are not the only driver of disease outbreaks, unconstrained mobility would have significantly accelerated the spreading of COVID-19 [35]. Zlojutro confirmed and evaluated the effectiveness of border control on 2009 H1N1 influenza control based on a global air travel network [36]. Another study in the journal *Science* suggested that travel restrictions have slowed, but not halted, the spread of the pandemic, and these restrictions violated international law as well [37].

How to carry out disease control at the national gate? Some scholars believe that the spread of animal and plant diseases is one of the most important side effects of the global economic integration process, the market supply for disease prevention and control is insufficient and there is also a lack of effective prevention and control of the negative externalities of disease generated by international trade [38]. Other scholars have called for the U.S. to expand disease detection capabilities at border crossings in the context of the expanding international wildlife trade, which now focuses on “protection” rather than “control” [39]. For the prevention and control of inbound pests and infectious diseases, some studies have concluded that the spread of infectious diseases in agricultural imports is an ongoing global hazard through a case study of the importation of fresh products carrying salmonella [40]. There are also some representative studies by Chinese scholars as follows. Danzi Liao proposes to establish a precise governance model [22]. Huang Binzhi proposed international collaboration between customs, port-related departments and inbound and outbound enterprises and passengers [41]. Yang Kunyu et al. explored the planning, monitoring and emergency response capabilities for the prevention and control of acute infectious diseases at ports [42]. Another study found that the proportion of detected cases of infectious diseases among people entering and leaving Chinese ports in countries along the Belt and Road increased from 9.9% in 2012 to 33.6% in 2016 and therefore explored strategies to reduce the potential risk of the transnational transmission of infectious diseases [18]. For the COVID-19 outbreak, Yang used the Susceptible Exposed Infected Recovery (SEIR) mathematical model of infectious diseases and the Government Response Stringency Index developed by the University of Oxford to assess the risk of importation of new coronavirus diseases from 14 bordering countries to China and they suggested that the capacity of port cities to deal with imported risks should be sorted out [43].

How does vaccination affect port security? Medical professionals, public policymakers and civil society view the issue very differently. There is currently a lack of generally convincing scientific empirical research on the impact of vaccines on the slowing of the spread of COVID-19 and on the probability of COVID-19 infection. Some scholars found that vaccination could reduce the disease severity of COVID-19 by the evidence that countries without universal vaccination policies have been more severely affected by COVID-19, compared to countries with universal and long-standing vaccination policies [44]. However, there is also “hesitancy” on the vaccination of COVID-19 and a lack of confidence in vaccines for COVID-19 could lower vaccination coverage rates, hence could pose security threats [45]. Also, most studies found that vaccine safety, efficacy, and potential side effects are top reasons for COVID-19 vaccination hesitancy among health workers [46]. In 2021, China’s COVID-19 vaccination has achieved the number of 2.1 billion [47]. Nevertheless, studies on the effects of vaccination on China’s port security or national security remain scarce.

Existing studies have analyzed the prevention and control of infectious diseases at ports from various perspectives. However, there are still shortcomings in the existing studies: (1) there are few specialized, systematic, or pivotal theoretical analyses around the “security of infectious diseases at ports”, and the research on how cross-border infectious diseases are “secured” and evolve into a fundamental global security issue is still relatively preliminary. (2) What are the deep-rooted structural constraints on building capacity for infectious disease security at ports? Most of the studies have analyzed the entries and exits of technical quarantine enforcement. However, the comprehensive analysis of the performance chain, legal basis, technological means, and international cooperation in controlling infectious diseases at ports is obviously inadequate. (3) How to improve the comprehensive governance capacity? Most of the studies have been conducted from various angles of the administrative system, policy process, technical means and the microprocesses of emergency response. Nevertheless, the analysis based on the global opening, cross-border factors, increased international tensions and China’s participation in global governance from the mid-macro perspective is still insufficient. This paper plans to examine the three issues in more depth.

## 4. The Deficient Capacity in China’s Infectious Disease Prevention and Control at Ports

During the 13th Five-Year Plan period (2016–2020), China’s 275 open ports have met the core capacity standards of the International Health Regulations (2005) for infectious diseases. In the overall deployment to combat the COVID-19 epidemic, China has established a joint prevention and control mechanism for port risk control [48]. However, the port infectious disease prevention and control capabilities are still deficient in many aspects.

### 4.1. There Are Obstacles in the Performance of the Functions of the Whole Chain of Government Departments

On the one hand, the pursuit of facilitation of customs clearance has overly squeezed the goal of “port security”. Ensuring customs clearance security and improving trade facilitation are the two basic principles for countries to build a modern customs system set by the “SAFE Framework of Standards for Global Trade Security and Facilitation” established by the World Customs Organization (WCO) in 2005. It is also the basis of the construction for China’s “border protection”, “security of the supply chain” and “port integration” [49]. In 2015, China accepted the Agreement on Trade Facilitation of WTO, demonstrating China’s determination to accelerate trade facilitation and expand openness. To fulfill the promise, the State Council of China has repeatedly called for reducing the time and cost of customs clearance at ports. Meanwhile, it also emphasized the need to resolutely prevent foreign infectious diseases, illegally traded waste and inferior products from entering China at the port to protect the safety of domestic citizens. The goals of facilitation and security are both necessary. However, in reality, the dual pressures of falling foreign trade and improving economic quality and efficiency have made the goal of trade facilitation a dominant position and substantially squeezed the implementation of port security functions. Risks and threats in the entry and exit chain are more concentrated at the “port”. In addition, new forms of international trade continue to emerge, such as cross-border e-commerce, service trade and technology trade. Therefore, new cross-border means of transport have emerged, such as oil (gas) pipelines and power transmission. Because of this, the port of entry and exit face more complex and unpredictable risks, such as trade frictions, terrorism, transnational crimes and data leakage in digital trade. However, the prospective attention and research on these new security issues are not enough and the systematic strategic planning and consequent policy measures are not sufficiently stocked.

On the other hand, there are obstacles to cross-sectoral cooperation. Major public health emergencies such as SARS and COVID-19 have demonstrated the urgent need for cross-sectoral collaboration. However, the inherent sectoral and compartmentalized system of the administrative system is incompatible with the integrated governance of cross-border comprehensive port security issues, which makes it difficult to carry out substantive coordination and cooperation between departments. Quarantine of infectious diseases at ports is only to prevent and control cross-border infectious diseases at the node of entry and exit. In fact, the prevention and control of cross-border infectious diseases is a complete chain involving the Customs, Maritime Affairs, Public Security, Border Defense, Immigrant, Health, Environmental Protection public departments, and some others. 

However, these departments are always criticized as “the railway police, each in charge of one section”, which has obvious fragmentation problems [50]. Additionally, as the statutory authority in charge of entry and exit ports, the General Administration of Customs overemphasizes vertical management and ignores horizontal coordination in the process of cooperating with the related responsible departments, resulting in the coexistence of overlapping responsibilities and disjointed management and the difficulty of vertical command and horizontal coordination, as well as the difficulty of information-sharing and mutual assistance in law enforcement.

### 4.2. Inconsistent Laws and Regulations

First, the internal laws of the port health and quarantine administrative system are not perfect or do not converge with each other. After several adjustments to China’s port health and quarantine agencies, in 2018, the Department of Entry-Exit Inspection and Quarantine, formerly under the General Administration of Quality Supervision, was transferred to the reorganized New Customs Administration, which is now the legitimate authority of the port health and quarantine. This means the integration of two separate systems of responsibilities. (The statutory duties of the new Customs expanded to cover the duties of quarantine supervision in the entry and exit process, including the formulation of the system of entry and exit health quarantine supervision, port health inspection and quarantine, epidemic monitoring, health supervision, port core capacity construction, port response to public health emergencies, etc. Meanwhile, Customs has always been the statutory public authority in charge of ports across the country. Available online: http://www.customs.gov.cn/ (accessed on 28 May 2022).) 

Accordingly, port health and quarantine have undergone corresponding changes in the central responsible units, legislation enforcement bodies, and supervision process, which has also brought about corresponding legal and regulatory problems: (1) As the “general law” of the New Customs Administration, the *Customs Law of the People’s Republic of China* (implemented in 1987 and 2017 Amendment, hereinafter referred to as the *Customs Law 1987*) is still positioned as the “promotion of economic and service trade” before the merger of the agencies. Therefore, the provisions on “port security” (such as border protection, anti-terrorism, health and quarantine supervision, emergency response, etc.) have become very unclear or even absent. (2) *The Frontier Health and Quarantine Law of the People’s Republic of China* (enacted in 1986 and 2019 Amendment, hereinafter referred to as the *FHQ 1986*) is the primary law governing port health and quarantine in China. However, the *FHQ 1986* and its Detailed Rules for the Implementation is at a disconnect with the COVID-19 response at ports of entry. At the same time, the *Customs Law 1987* and the *FHQ 1986* are not connected and coordinated in the core content of port health and quarantine procedures and supervision and are unable to respond effectively to the current more complex port health inspection and supervision requirements. (3) There is a significant ambiguity in the provisions of the responsible departments of port infectious disease emergency response and port core capacity construction and the law enforcement discretion. Thus, the legal deterrent to law enforcement objects is very insufficient. What is particularly important is that there is a lack of legal basis for emergency control measures for port disease control under relevant extreme conditions.

Second, the “port health and quarantine” related laws do not converge, and they lack an overarching law. There are dozens of laws and regulations for responding to infectious disease emergencies at ports (Table 2). However, there are still two obvious problems: (1) the general public health emergency law basically does not mention the content of “health quarantine at ports”, and there are no supporting regulations proposed to dovetail with the provisions of the *FHQ 1986*, which leads to the fact that although the Customs Administration serves as the “legitimate authority” of the port health quarantine, they are faced with the dilemma that their responsibilities and departmental cooperation “are not based on laws”. (2) Although there are special laws (e.g., *The FHQ 1986*), “departmental laws” (e.g., the *Customs Law 1987*) and “general laws” (e.g., the *Emergency Response Law 2007, Law on Prevention and Treatment of Infectious Diseases 1989*), there is a lack of a “general law” that can govern the relationship between all parties, clarify the responsibilities of all parties and their mutual cooperation.

Third, there is a mismatch between domestic law and international law. Currently, the International Health Regulations 2005 (IHR 2005) is the only international health agreement to date that is legally binding on member states on the control of infectious diseases. For example, on 31 January 2020, the World Health Organization declared in accordance with the IHR 2005 that the outbreak of novel coronavirus pneumonia (2019-nCoV) in Wuhan, China constituted a Public Health Emergency of International Concern (PHEIC). In the international spread of diseases, in order to ensure maximum safety and minimize interference to international traffic operations, this law addresses key issues such as the responsibilities of the international focal point of each member state, the notification procedures and handling measures for PHEIC incidents and the building of core port capabilities. It is also the international legal basis for domestic health quarantine in member states.

The new version of IHR 2005 implemented on 15 June 2007 [51] incorporates several new points, including firstly, the addition of “Public Health Emergency of International Concern” (referred to as PHEIC), which means a special event that is determined, as provided in these regulations: (1) to constitute a public health risk to other states through the international spread of disease and (2) to potentially require a coordinated international response. Secondly, it extends the prevention, protection and control of international transmission of “disease” to a broader range of public health management and control of biological, chemical, nuclear and radiological hazards. It places greater demands on member states regarding their capacity, responsibilities and obligations to respond to public health emergencies.

In May 2007, the Chinese government issued a statement that IHR 2005 applies to the entire territory of China, which means that it is the source and basis for the revision of China’s national health and quarantine law. However, the global COVID-19 pandemic has exposed two aspects of the mismatch between the health and quarantine regulations at China’s port and IHR 2005: (1) The *FHQ 1986* that China is currently implementing was implemented in December 1986. It has been 36 years since then, and it is still based on the old version of the International Health Regulations (1969) as the actual background and legislative basis. Although it has been revised three times, its main content, its implementation rules, and related laws all lack the legal items of PHEIC, and only have corresponding clauses for “quarantine infectious diseases”, “monitoring infectious diseases”, and “infected areas”. (2) Although the current *FHQ 1986* attempts to expand the concept of “quarantine” to include all health measures taken to prevent/control the occurrence, transmission and spread of public health risks, such as epidemic monitoring, early warning, notification, health education, vaccination, entry and exit inspections of personnel and traffic vehicles, the building of core port capabilities, etc., have not achieved practical results. This has resulted in an insufficient basis for front-line enforcement at Chinese ports and a lack of targeted legal basis for responding to PHEIC, far from meeting the needs of the IHR 2005 governance system of “broader public health management and disposal of biological, chemical, and nuclear, radiological hazards” [28]. Similarly, the WTO’s security exceptions provide that WTO obligations are not to be fulfilled in “exceptional circumstances”, and there is no clear answer as to whether the WTO has the authority to rule on trade disputes arising from national security measures [52].

In addition, there is a phenomenon of “situational law enforcement” based on “experience” and “discretionary” in the process of law enforcement of the prevention and control of infectious diseases at the port. For example, the determination of whether an incoming person is suspected of being infected with the disease, whether the transportation requires sanitary treatment and whether taking emergency fever-reducing medicine constitutes evasion of quarantine or resistance to quarantine can be either ignored or strictly enforced on the spot, while the corresponding laws and regulations are not well set up and lack of details. Although the General Administration of Customs and five other departments have jointly established six types of behaviors that can be convicted or investigated for obstructing the sanitary and quarantine at port [53], they only set penalties for evading quarantine or causing the spread of the epidemic and do not make explicit provisions for uncooperative behaviors, which are not enough to deter disruption of sanitary and quarantine enforcement.

### 4.3. Shortage of Key Technologies

International practices in public health governance have shown that technology plays a key role in helping to control the spread of disease. In response to the COVID-19 outbreak, China created innovations and used digital technology on a large scale, including nationwide health codes, drones, telemedicine powered by 5G technology and smartphone-automated calls, etc. In addition, there is also the research and development and application of AI technology supported by 5G, for example, for temperature measurement and recording, close contact tracking, intelligent patrol, AI medical diagnosis, unmanned material delivery and many other aspects. A CNN article lists some of the new technologies used in China’s fight against the epidemic, including drones, disinfection robots and supercomputers, and says that this is a test of China’s technological capabilities in the pandemic [54]. An article in the *Harvard Business Review* by MIT Professor Yashang Huang analyzes how China’s epidemic control technology has helped the accurate close contact tracing in the fight against COVID-19 [55].

However, China’s use of smart technology to fight the epidemic is not without problems. On the one hand, the key scientific and technological support for China’s fight against the epidemic is the smart technology supported by big data, information technology and 5G. China has made full use of data resources and achieved effective data governance in epidemic prevention and control. A large number of research studies and news reports have confirmed the benefits of big data in data acquisition. Nevertheless, data tech application still faces problems such as cross-border data connectivity in epidemic prevention and control at ports. Among them, more specific deficiencies include the collection and application of big data at ports, biological laboratories, cross-departmental data sharing and quarantine equipment: (1) the port quarantine departments have insufficient data mining and analysis and application capabilities in identifying and responding to significant outbreaks of infectious diseases, and there is a lack of institutions, and medical resources capacity of county-level medical institutions are severely limited. “Medical resources at ports in China are severely inadequate in the South-west China” [56]. There is also an insufficient number of key laboratories and even fewer laboratories reaching biosafety level 3. The National Health Commission of China also points out that the early warning system at ports still needs to be improved [57]. (2) The port-related departments stock real-time full-caliber import and export data and a large amount of information shared by various related departments, but the data is dormant, and data value mining and utilization is far from sufficient, especially the early intervention for potential risk monitoring is insufficient. In addition, the acquisition of big data on epidemic diseases and epidemics is slow, and early warning monitoring technology is inadequate. (3) Information on the spread of cross-border epidemics has a single source and low credibility. At the same time, the means of monitoring and early warning are relatively simple, and the manual extraction method for epidemic monitoring is time-consuming and labor-intensive; risk assessment lacks quantifiable indicators, and the assessment results are not included in the customs clearance process of the port department as a risk control instruction. Additionally, the risk assessment capabilities are relatively inadequate, and in particular, the special risk assessment capability of PHIEC, which was newly established by IHR 2005, is very weak. (4) The introduction and development of new technologies for vector biology monitoring and health supervision are scarce, and CT intelligent map review has not been effectively used, as well as the development progress of non-intrusive inspection and intelligent drawing inspection systems represented by “intelligent machine inspection” is relatively slow. Problems of insufficient accuracy remain in areas such as elimination and logistics. In July 2022, China’s state council pointed out that the “disinfection of imported goods still needs to be improved” [58]. Inspection facilities and equipment and cargo quarantine facilities and equipment are insufficient. As the Chinese official points out, the planning and building of the infrastructures of China’s ports are still limited [59].

On the other hand, the relatively macro-level problem is that China’s technology for accurately locating close-contact people mainly relies on facial recognition, geographic location tracking and the integration of local government big data with strong administrative leadership. The relative laxity of the Chinese social and legal environment on personal information protection has contributed significantly to this. However, face recognition and precise positioning involve issues such as identity information collection and personal privacy protection. The excessive free collection and application of big data has not only aroused a certain degree of public resistance to big data’s anti-epidemic but also virtually expanded the government’s authority. On the whole, it has to be said that this has brought challenges to the good governance and civilization of “technological anti-epidemic”.

### 4.4. Challenges in International Cooperation

Firstly, China vigorously develops the Belt and Road Initiative and actively conducts health and quarantine cooperation with countries in the region. However, countries in the Belt and Road region have enormous difficulties of their own in terms of international cooperation in cross-border infectious diseases. Overall, many countries in this region are developing countries and face problems of poverty, unemployment, insufficient energy, ethnic conflict, etc. Therefore, comparatively speaking, these countries have always shown a lack of concern and strength in public health issues, resulting in insufficient public medical investment, outdated medical infrastructure, low medical standards, non-standard industry management, and weak willingness for international cooperation. Therefore, they lack basic public health conditions and comprehensive national strength in the face of COVID-19 challenges. This situation actually makes it more difficult for them to cooperate with China in cross-border infectious disease response.

Secondly, international cooperation between China and its neighboring countries is also challenging. For example, some South Asian countries such as India experience barriers such as poor health knowledge among their residents, undeveloped primary medical conditions, and less access to modern medicine. Then there are Central Asian countries, such as Kazakhstan, where the Ministry of Health was abolished several times from 1997 to 2002 due to historical, political, and geopolitical factors. The principal leadership frequently changed, with constant riots and internal conflicts. Political instability, lack of medical resources, limited testing capacity, and non-compliance with epidemic prevention regulations are the main difficulties that epidemic prevention faces in this region. In addition, the frequent internal armed conflicts and natural disasters in many countries have damaged and weakened the public health system, leading to a decrease in the accessibility of health services, which has directly caused more difficulty in terms of health quarantine in China.

Thirdly, in the process of fighting against COVID-19, the fragility of public health capacity across the West has been exposed, and the leadership and cooperation of global powers are seriously lacking. Richard N. Haass, president of the Council on Foreign Relations of the United States, wrote in *Foreign Affairs* magazine that the pandemic crisis has shown that U.S. leadership has weakened, global cooperation has faltered, and great power discord. Besides, the WHO’s inability to lead great power cooperation also indicates the lack of global cohesion. In addition, China and the United States, the two most influential countries in the world, have failed in their joint response to the epidemic. The entire international community lacks not only a consensus on the crisis but also lacks due leadership of significant powers and coordination among countries [60].

At the same time, COVID-19 has also exposed the international community’s unstable long-term mechanism for mutual assistance in international law enforcement around public health emergencies. The international community has a basic consensus on strengthening cooperation at ports regarding technology, statistics, counter-terrorism, intellectual property protection, Authorized Economic Operators (AEO), data exchange, trade facilitation, and other aspects. However, cooperation in these areas also has the shortcomings of being mainly bilateral and mostly limited to paper agreements. An extensive and stable international law enforcement mutual assistance mechanism has not been established. More prominent problems include poor data interaction and communication channels, long information exchange cycles, and especially narrow intelligence exchanges. This has led to difficulties forming a genuinely effective bilateral or multilateral mechanism in crucial areas such as port health and quarantine standards and information sharing.

In addition, the status of compliance with international conventions related to transnational infectious disease prevention and control is problematic. For example, international shipping hubs, with many outgoing and incoming sea vessels, interlocking routes, and large and complex interactions of seafarers and passengers, are sensitive channels that contribute to the transnational spread of infectious diseases. According to statistics, there is a total of about 1,647,000 seafarers on the global international trade merchant ships, 900,000 of them from developing countries. Six of the world’s seven international routes pass through the port of Zhoushan in China, and as of the end of 2020, China has a total of 1.716 million registered crew members, including nearly 593,000 seafarers on international voyages [61]. 

After the outbreak of the COVID-19 pandemic, there have been incidents on the American Carnival Group’s “Grand Princess” and “Diamond Princess”, as well as Panama-registered “Grand Progress” and several international cruise ships. Cruise ships are difficult to dock, seafarers cannot change shifts and are infected with COVID-19, and vaccinations and treatment are insufficient. It again reflects the relevant issues of international conventions. For example, the Maritime Labour Convention (MLC) stipulates provisions for seafarer workers on shore leave, annual leave, and repatriation after a maximum of 11 months of service. However, despite the 2021 International Transport Forum (ITF) summit held in the midst of the COVID-19 crisis, the framework of the agreement on crew changes developed by the International Chamber of Shipping (ICS) and the International Maritime Organization (IMO) has failed to persuade membership country to follow them. According to statistics, only 25% of seafarers worldwide will be able to change shifts properly from March to August in 2020, and 250,000 crew members were overdue in August. International organizations such as the WHO and IMO also require that seafarers be vaccinated, yet less than 1/3 of member states follow the directive. Similarly, the security exception clause in the WTO also stipulates that member states should not be required to perform the obligations in “special circumstances”, and there is no clear answer to whether the WTO has the right and how to adjudicate trade disputes caused by national security measures of various countries. In fact, it creates a “legal basis” for “special circumstances” for transnational non-cooperation in infectious diseases [52].

## 5. Reflection and Discussion

In terms of fighting against the COVID-19 crisis, compared with the laissez-faire style of epidemic prevention and control adopted by most countries in the world, China has adopted almost a quasi-war style of strict prevention and control as domestic policies as well as policies at ports of entry and exit. China’s epidemic control at all costs has been effective in reducing the number of confirmed cases and deaths and has thus completely avoided the possibility of societal disorder. Evidence has shown that if Omicron spread free in China, it would cause a number of 1.06 million death cases, and elevating vaccination rates and improving vaccination effectiveness would both contribute to alleviating this situation [62]. However, China’s success in combating the epidemic does not mean that all prevention and control initiatives and implementation were fully justified. The initial analysis also shows that there are still specific measures that deserve reflection and improvement and the same goes for port responses.

First, the laws on the prevention and control of infectious diseases at ports need to be improved. The preceding analysis reveals at least the following aspects: (1) *The Customs Law of PRC* (the *Customs Law 1987*) needs to add new statutory functions such as “border protection, anti-terrorism, promotion of trade security and convenience, and port security”. In other words, the *Customs Law 1987* requires positioning customs based on “promoting economy and trade” and “ensuring security” simultaneously. In addition, the law takes the port security function as a guide, re-plans the layout of the existing customs laws and regulations, revises and improves lagging statutes and rules, resolves conflicts between higher and lower laws, expands the coverage of provisions, enhances operability, rationalizes overlapping responsibilities, improves the directory system, and clarifies the legal status and specific duties of customs in the port security function. (2) Given the gap in the public health emergency-related laws on “health quarantine at ports”, it is necessary to collate the contents of the 1986 *Frontier Health and Quarantine Law* (the *FHQ 1986)* and the existing public health emergency-related laws and clarify the specific responsibilities and joint response mechanisms of all relevant departments such as customs, health, security, border defense, civil aviation and transportation in “health quarantine at border ports”. (3) In response to the lack of convergence between the relevant domestic laws and the specific provisions of IHR 2005, the Chinese Law on *FHQ 1986* should be amended to add the entry of *Public Health Emergency of International Concern (PHEIC)*. At the same time, the *FHQ 1986* should expand the concept of “quarantine” to better correspond to the requirements of the biological, chemical, nuclear and radiological hazards caused by a broader range of public health management and disposal, the principles of which were stabled by IHR 2005.

Second, the risk prevention, control and disposal capacity at ports throughout the whole chain needs to be further strengthened. The scope of port health quarantine and supervision should be expanded from the port gate of import and export to the entire chain of element flow, ensure prevention and control in advance, monitor the risk of each node in the whole process, and emphasize the prevention and control of the source of risk, make faster, earlier, and more accurate analysis and position risk points and create a whole chain risk prevention and control system extending to the source and terminal. For example, the construction of overseas infectious disease surveillance sentinel sites can be promoted. In 2016, when the first imported Zika virus case was detected in China’s port, the person entering China did not have fever symptoms at the port but was only detected through surveillance at overseas surveillance sentinel sites. As a matter of fact, of the 24 Zika virus cases seen throughout the year, only half were found at the port site [63].

Third, the infectious disease surveillance capacity at ports should be empowered by innovative technology. The COVID-19 crisis made the conventional technical means of port health quarantine seem obsolete. For example, the survey showed that nearly 77% of inbound infectious disease cases in the 272 ports in 2014–2016 were detected through infrared body temperature monitoring [63]. However, infrared body temperature surveillance receipts are often disturbed by variable factors such as weather and alcohol consumption. It is challenging to detect infected individuals in the incubation period of disease onset or asymptomatic—it appears even more so in the COVID-19 scenario. Therefore, it is essential to promote intelligent technologies’ development, innovation and application. (1) Accelerate the intelligence of infectious disease control at ports based on digital technologies such as big data, artificial intelligence, cloud computing and the Internet of Things. Improve laboratory construction, equipment configuration, monitoring technology, platform sharing and cooperation, complete the port laboratory and build a three-dimensional monitoring system for comprehensive monitoring, prevention, and control of nuclear, biological and chemical harmful factors. (2) Strengthen the application of big data analysis, extensively collect disease risk intelligence, unify data standards, construct suitable risk analysis “data pool”, strengthen information early warning and intelligent perception/identification and improve regulatory accuracy and effectiveness. (3) Promote port-related departments by investing in information technology equipment, improving the automatic data collection function and building a cross-sector compatible information system, such as real-time data transmission. Promote the research and application of 5G-based intelligent ports and realize the integration of 5G technology and port infectious disease prevention and control.

Fourth, deepen transnational collaborative governance. (1) On the basis of multilateral and bilateral trade facilitation with countries along the “Belt and Road”, China can strengthen the mutual recognition of supervision, information exchange, and mutual assistance in law enforcement with these countries in the joint control of infectious diseases at ports, and based on this, jointly strengthen epidemic risk management ability. (2) Cooperate with international organizations in the management of infectious diseases. Based on multilateral frameworks such as the World Customs Organization (WCO), the World Health Organization (WHO), the World Trade Organization (WTO), the World Organization for Animal Health (OIE) and the International Plant Protection Convention (IPPC), China can work with member states to improve mechanisms such as port nuclear, biochemical risk prevention and control, disease monitoring and emergency response and joint response to transnational epidemics. (3) During the 13th Five-Year Plan (2016–2020), a long-term mechanism for bilateral cooperation at ports between China–Russia, China–Kazakhstan, China–Mongolia and China–Vietnam was established. Based on these bilateral cooperation frameworks, China should further promote multilateral and regional mechanisms such as the Greater Mekong Subregion (GMS), the Shanghai Cooperation Organization, the Lancang-Mekong Cooperation, and the Greater Tumen Initiative (GTI) and expand multilateral or bilateral cooperation with more neighboring countries in information exchange and emergency response for port health and quarantine. (4) Exchange information related to translational infectious disease prevention and control with the world, including vaccine development, infectious disease information systems, disease surveillance technologies, etc. For example, the China–Africa Health Silk Road, the World Health Forum, the Boao Forum for Asia Global Health Forum, the Global Health Security Agenda (GHSA), etc., are international platforms for exchanging international media for disease prevention and control.

## Figures and Tables

**Table 1 healthcare-10-01359-t001:** Health Quarantine at Border of China (2015–2018).

Year	Arrivals and Departures (10,000 Person-Times)	Persons with Symptoms of Infectious Diseases (Person-Times)	Confirmed Infectious Diseases (Cases)
2015	49,100	101,000	15,500
2016	55,100	141,600	25,505
2017	57,500	122,400	27,600
2018	65,100	123,500	27,000
Total	226,800	488,500	95,605

Source: General Administration of Customs and CDC of China.

**Table 2 healthcare-10-01359-t002:** Laws on Port Health Quarantine of PRC (International and Domestic laws/Regulations not Included).

No.	Title	Year of Enforcement/Amendment
1	*Law of the People’s Republic of China on Prevention and Treatment of Infectious Diseases*	1989; 2004 Amendment; 2013 Amendment
2	*Emergency Response Law of the People’s Republic of China*	2007
3	*Law of the People’s Republic of China on the Promotion of Basic Medical and Health Care*	2019
4	*Frontier Health and Quarantine Law of the People’s Republic of China*	1986; 2007 Amendment; 2009 Amendment; 2018 Amendment
5	*Wild Animal Conservation Law of the People’s Republic of China*	1988; 2004 Amendment; 2009 Amendment; 2018 Amendment
6	*Animal Epidemic Prevention Law of the People’s Republic of China*	1997; 2007 Amendment; 2013 Amendment; 2015 Amendment; 2021 Amendment
7	*Civil Code of the People’s Republic of China*	2020
8	*Public Security Administration Punishments Law of the People’s Republic of China*	2005; 2012 Amendment
9	*Law of the People’s Republic of China on Administrative Penalty*	1996; 2009 Amendment; 2017 Amendment; 2021 Amendment
10	*Criminal Law of the People’s Republic of China*	1979; 2020 11th Amendment
11	*Decision of the Standing Committee of the National People’s Congress to Comprehensively Prohibit the Illegal Trade of Wild Animals, Break the Bad Habit of Excessive Consumption of Wild Animals and Effectively Secure the Life and Health of the People*	2020
12	*Biosecurity Law of the People’s Republic of China*	2020
13	*Law of the People’s Republic of China on the Prevention and Treatment of Infectious Diseases and its Measures for Implementation*	1991

## Data Availability

The data used to support the findings of this study are available from the corresponding author upon request.

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
