# Peer review of "United by Contagion: How Can China Improve Its Capabilities of Port Infectious Disease Prevention and Control?"

_healthcare, 2022, doi:10.3390/healthcare10081359_

Round 1
Reviewer 1 Report
Dear authors,
It is an in-time article that reviews the China’s port security of infectious disease (Section 1-3). Suggestions were given in the aspects of infrastructure of Chinese government, law, international partnership, and technologies (Section 4-5). Some improvement could be made to the article:
1. Vaccination is a major preventive measure for COVID-19. Vaccination can not only reduce disease severity of COVID-19, but also reduce viral load of the infected persons and thus the chance of disease spreading from the port. The policy of Chinese government in implementing vaccination scheme and whether vaccination contribute to port security should be discussed in the current paper. Whether and how could the vaccination policy at port security be improved can be discussed on top.
2. For section 4.3 (line 480-513), more evidence is needed to support the claim “the use of key technologies in the prevention and control of infectious diseases at Chines parts is still inadequate”. For example, what precisely mean “slow” in line 496. What precisely mean “low” and “inadequate” in respectively line 498 and 499. Also, what precisely mean “insufficient number of key laboratories/Biosafety level 3 laboratory” in line 508, i.e. how much would be sufficient?
3. It will be better to provide a conclusion section at the end of this article to precisely summarize the claim for how China can improve port infectious disease control.
4. Line 296-297, better to write “ …fresh product carrying salmonella.”
5. Line 574-576 and line 585-586 can be re-written for grammatical correction.
Thank you for your attention.
Reviewer 2 Report
Dear authors,
Your work is very interesting; this analyzes many aspects of internal policy and control of infectious diseases. You have created a complete history of the development of many infectious diseases and analyzed many essential factors. Moreover, you have performed a very detailed, rigorous analysis.
I have a few considerations for you:
– Your paper looks more like a review or newspaper article than a research paper. Furthermore, there are no data or statistical analysis citations to confirm your thesis. In this form, although the article is interesting, I believe it is more suitable for publication as a simple review. I recommend that you specify it with the authors for greater clarity.
– 2nd and 3rd Paragraphs: I think these paragraphs are too long and dispersed. It is undoubtedly essential to describe in detail a topic to better explain its various facets, but the article is very long and risks losing the reader's attention while reading. I suggest you shorten and reduce some parts, especially in the second paragraph.
– In the introduction, you did a good work describing the infectious diseases that have spread due to migration and political choices. I recommend an interesting article about it, helpful to include in your references: G. Troiano, N. Nante. Political and medical role in the last Ebola outbreak. J Prev Med Hyg. 2017 Sep;58(3):E201-E202.
– In the second paragraph, from lines 189 to 207, you imply judgments on foreign political decisions. A scientific article is not a newspaper; objective information must be reported on the documented facts and present data without suggesting any judgment. I invite you to review this part and refrain from judgments to avoid losing credibility.
– In the final part of the second paragraph, you list the initiatives taken by China to prevent infections at ports, but insert a number in brackets to number them. Again, I suggest you create a bulleted list to make reading better.
– Regarding table 1, for better reading, I suggest you insert and put in brackets the percentages of the numbers in the second and third columns.
– In the discussion paragraph, on lines 598 and 599, insert some data considering them to be effective and specific, without inserting citations. I recommend entering the articles you are referring to validate your statements.
- I suggest you review the final paper because there have been some problems, especially in punctuation: for example, you have inserted the "semicolon (;)" inappropriately in the abstract (line 20 and line 22) and one point too many on line 28 in the first paragraph of the text.
